# Design and Development of an Ultraviolet All-Sky Imaging System

**DOI:** 10.3390/s23177343

**Published:** 2023-08-23

**Authors:** Thasshwin Mathanlal, Javier Martin-Torres

**Affiliations:** 1Department of Planetary Sciences, School of Geosciences, University of Aberdeen, Meston Building, Aberdeen AB24 3UE, UK; javier.martin-torres@abdn.ac.uk; 2Instituto Andaluz de Ciencias de la Tierra (CSIC-UGR), 18100 Granada, Spain

**Keywords:** all-sky camera, UV imaging, sensor, validation, COTS

## Abstract

All-sky cameras capture a panoramic view of the full sky from horizon to horizon to generate a wide-angle image of the observable sky. State-of-the-art all-sky imagers are limited to imaging in the visible and infrared spectrum and cannot image in the UV spectrum. This article describes the development of an all-sky imaging system capable of capturing 130° wide-angle sky images from horizon to horizon in the UV-AB spectrum. The design of the UV all-sky imaging system is based on low-cost, accessible, and scalable components to develop multiple images that can be deployed over a wider geographical area. The spectral response of the camera system has been validated in the UV spectrum (280–420 nm) using a monochromatic UV beam with an average power output of 22 nW. UV all-sky imaging systems complement existing infrared and visible all-sky cameras. They have wide applications in astronomy, meteorology, atmospheric science, vulcanology, meteors and auroral monitoring, and the defence sector.

## 1. Introduction

All-sky imaging systems capture a panoramic view of the sky from horizon to horizon to generate a wide-angle image of the observable sky. American astronomer Edward Emerson Barnard (1857–1923) took the first panoramic image of the sky in 1889 when he combined 50 wide-angle photograph mosaics into a single panoramic image, which he published in his book *A Photographic Atlas of Selected Regions of the Milky Way* [1]. The first wide-angle photograph with a 180° field of view captured on a single image was accomplished 12 years later by his contemporary Robert Williams Wood (1868–1955) using a water-filled box and pinhole exploiting the differing refractive indexes of air and water. The image represented how a fish looking upward at the water’s surface could see the whole sky in a finite circle, and he termed it the fisheye effect [2], a term still used today. A more practical approach to all-sky imaging using a lens-based system was developed by Robin Hill (1899–1991) in 1924 using a simple setup of a large negative meniscus lens coupled with a pair of single elements behind it [3]. However, the lens suffered low relative apertures. Still, he could use it to take stereoscopic whole-sky images, recording cloud patterns in three dimensions. To mitigate the low relative apertures associated with fisheye lenses, reflecting catadioptric systems for all-sky imaging gained popularity. The use of a standard camera pointing over a convex mirror to capture the full sky was documented in 1949 in the Bulletin of the American Meteorological Society [4]. Reflecting all-sky camera systems do not suffer from chromatic dispersion associated with refracting lenses, eliminating the need for complex corrective optics. A reflecting catadioptric system to image the sky has been used since 1956 at the Swedish Institute of Space Physics (IRF) in Kiruna to study aurora borealis [5].

Michael Gadsden (1933–2003), at the University of Aberdeen, UK, developed a reflecting catadioptric all-sky imager in 1978, in which he utilised two convex mirrors with a standard camera lens to obtain a 180° field of view image, retaining the relative aperture of that of the standard lens [6]. Lower relative apertures of fisheye lenses prevalent in the market in the early 1950s were a limiting factor for night-sky imaging, and reflecting catadioptric systems remained popular [7]. With the advent of wider aperture fisheye lenses, such as the Nikkor 10.5 mm f/2.8 in Tokyo, Japan, all-sky imaging began shifting from reflecting systems to refracting systems.

All-sky imaging cameras find application in many disciplines, ranging from astronomy, meteorology, atmospheric science, vulcanology, auroral monitoring, and the defence sector. For example, all-sky images have been used to supplement the radiometric measurements of the distribution of diffuse solar radiation [8], monochromatic all-sky imaging has been used to study auroral precipitation patterns [9], or they have been applied in studying cloud macrophysical properties and their influence on the hydrological cycle and energy balance [10]. To date, all-sky imaging cameras operate in the visible and infrared regions, and an imaging system does not exist to provide all-sky images in the ultraviolet (UV) spectrum domain. The lack of a fisheye lens constructed out of UV-transparent materials has been a main limiting factor in the development of such wide-angle cameras in the UV spectrum. Circumventing the expensive production of a fisheye lens from UV-transparent materials, we have developed a UV all-sky imager designed using a catadioptric system. The designed all-sky UV imaging system can capture a 130° sky image in the UV-AB spectrum (280–400 nm). Catadioptric systems are noncentral imaging systems in which misalignments exist between the camera and the reflective mirrors, posing a challenge in calibrating the radiance and angular projections. There exist calibration algorithms, namely the blackbox model [11], the complete model [12], and the generalized unified model [13], to solve for the discrepancies in projection. This paper details the development of the imaging system and does not cover the radiance and angular calibration of the imaging system, which will be detailed in a separate article.

All-sky imaging in UV has wide applications in atmospheric science, pollution monitoring, and defence sectors. The all-sky camera operating in the UV-AB region is an excellent tool for studying the enhancement of solar radiation in the UV region due to radiation scattering from the clouds. An enhancement of 6% in the UV region from 295 to 385 nm was reported in a study at Puna de Atacama, Chile (23.3° S, 3700 m a.s.l) [14]. Solar UV irradiance in the UV-B spectrum was reported to be enhance due to cumulus clouds near the sun’s apparent position compared to cloudless sky conditions [15]. Similar studies of the enhancement of UV radiation due to scattering from clouds have been reported in [16,17,18,19,20]. The studies use radiance data from UV pyranometers and cloud-cover data from in situ imagers or satellite data to determine the type and position of clouds on the enhancement of solar irradiance in the UV spectrum. All-sky imaging in the UV spectrum can find application in the study of cirrostratus and noctilucent clouds due to their ability to scatter sunlight in the spectral range of 252–292 nm [21]. The all-sky imaging system can monitor sulphur dioxide emissions from anthropogenic activities, i.e., industries, or natural sources, i.e., volcanoes and forest fires. Sulphur dioxide (SO_2_) has a strong absorption in a narrow wavelength at 310 nm and a weak absorption at 330 nm [22]. Using narrowband filters at 310 nm and 330 nm in the all-sky imager, mapping sulphur dioxide emissions in a 130° field of view is possible. A low-cost smartphone sensor-based UV camera for volcanic SO_2_ emission measurements has been reported in [23]. However, the field of view is restricted to a maximum of 28.3°. UV all-sky imagers can find application in the study of aerosol optical depth by Mie scattering, which is essential for climate study and radiative balance [24]. They can find application in estimating the night-sky background in the UV spectrum. All-sky imaging in UV can also find applications in the defence sector for monitoring the sky, as imaging in UV produces a higher contrast of objects than in visible or infrared (IR) spectrum due to better UV reflection from tiny surface features. The UV all-sky imager is to be deployed in the Galileo project of Harvard University to study unexplained aerial phenomena (UAP) by investigating UV wavelengths as a part of the PArticle Counter k-index Magnetic ANomaly PACKMAN instrument [25].

The design, development, and validation of the all-sky UV imaging system are detailed in this paper. A commercial off-the-shelf (COTS)-based approach is used to design the system by (i) modifying a commercial monochrome camera sensor to improve its responsivity to the UV spectrum; (ii) designing a UV lens with commercially available fused silica elements and a commercial lens body; and (iii) designing a catadioptric reflecting all-sky imager using commercially available convex mirrors. Using a COTS-based approach reduces the cost of developing a network of imagers with multiple units deployed at multiple locations. Deploying such UV all-sky imagers over a broad spatial area can help understand the UV environment and its influence on the atmosphere, hydrosphere, lithosphere, and biosphere.

## 2. Construction

UV imaging of the full sky requires capturing images with an ultra-wide-angle lens. Visible all-sky imaging cameras have a fisheye lens with a field of view between 100° and 180°. The absence of an ultra-wide-angle lens made of UV-transparent materials, such as fused silica, quartz, calcium fluoride, and lithium fluoride, necessitates the use of a reflecting catadioptric system to image the full sky. UV-AB can penetrate the atmosphere and reach the earth’s surface; however, its magnitude depends on geographical parameters such as latitude, longitude, and altitude and local conditions, such as cloud cover, aerosol concentration, and atmospheric gas concentration. UV-A constitutes 90 to 99% of total UV reaching the earth, while UV-B accounts for 1–10% [26]. The high temporal and spatial variability in the UV-AB intensity reaching the earth’s surface warrants the need for UV-transparent materials to construct the all-sky imaging system to improve the UV signal response. The developed reflecting all-sky imaging technique allows a field of view of 130° with higher relative apertures of f/1.6 to capture most of the UV-AB light that has penetrated the earth’s atmosphere. In this section, we detail the construction of the UV all-sky imaging camera system, beginning with the adaptation of a commercial high-sensitivity camera sensor for UV imaging, followed by the development of a 22 mm focal length lens made of fused silica components and concluding with the design of the catadioptric system incorporating the above-mentioned camera and lens.

### 2.1. UV Camera Sensor

The core of the reflecting all-sky camera is the imaging sensor itself. Commercial UV-sensitive imaging sensors exist, but their cost and availability are major limiting factors. The recent commercial imaging sensor sensitive to the UV spectrum is the IMX487 sensor from Sony Corporation, Tokyo, Japan, which is currently in the prototyping phase for commercial markets [27], and costs more than GBP 10,000. However, a commercial imaging sensor with similar specifications capturing in visible spectrum, such as the Sony IMX546, costs only about GBP 1000 [28]. Such sensors find application in the material analysis field as imaging in UV produces a higher contrast of defects than is observable in the visible or IR spectrum due to reflection from tiny surface features. To mitigate the high cost of procuring a UV-sensitive image sensor, we modified a commercial charge-coupled device (CCD) sensor used for deep-sky imaging by improving its sensitivity in the UV spectrum. CCD backside-illuminated sensors have higher UV quantum efficiency than front-illuminated sensors [29]. We chose the ICX205AL sensor (Sony Corporation, Tokyo, Japan) due to its low thermal noise and high-speed electronic shutter, capable of matching the higher apertures needed for all-sky imaging. The ICX205AL sensor belongs to the family of HAD (Hole accumulation diode) sensors, which offer high sensitivity and low dark current required for deep-sky imaging. The ICX205AL sensor has demonstrated application in night all-sky imaging characterizing atmospheric turbulence [30,31]. The high sensitivity of ICX205AL in such demanding low-signal imaging requirements justified its application in UV camera development. We utilised the DMK 41AU02.AS Monochrome Camera from Imaging Source Europe GmbH, Bremen. Germany with an ICX205AL sensor. The sensor has a quantum efficiency of 40% at 400 nm [32]. The specifications of the ICX205AL sensor are listed in Table 1.

Monochrome sensors do not have the colour filter array (CFA), which generates colour information for each pixel. The dyes used in the CFA reduce sensitivity in the UV spectrum rapidly from 400 nm to around 350 nm [33]. Hence, the monochrome camera has, in general, a higher sensitivity to UV compared to colour cameras. With no CFA in monochrome ICX205AL, the only element on the image sensor that can reduce UV sensitivity is the standard sensor cover glass. The standard cover glass ICX205AL sensor was removed and replaced with a fused silica cover glass ICX205AL sensor to ensure maximum UV transmissibility. The replacement of the fused silica cover glass on the ICX205AL sensor was performed in an argon atmosphere by Eureca Messtechnik GmbH, Cologne, Germany. Figure 1 shows the replacement of the standard ICX205AL sensor with the fused silica cover glass ICX205AL sensor on the DMK 41AU02.AS Monochrome Camera board. We tested the sensor’s UV response before and after the cover glass change. The details of these tests are presented in the results and discussion section.

### 2.2. UV-Transparent Lens

Catadioptric systems use reflecting (mirror) and refracting (lens) elements for imaging. Most commercial lenses available in the market are made of optical glass, with excellent transmittance in the visible range of 400 nm to 800 nm. The availability of lenses transparent to the UV spectrum is limited and costly and is a major limiting factor in developing a commercial all-sky imager sensitive to the UV spectrum. An example of a lens that can pass UV is the Jenoptik 60 mm F/4 Apochromatic (APO) lens with a wavelength passband between 290 and 1500 nm that costs around GBP 7000 [34]. An alternative to the Jenoptik 60 mm F/4 Apochromatic lens is the Nikkor 105 mm F/4.5, which has a wavelength passband between 220 and 900 nm. These lenses are unique, in high demand, and cost around GBP 12,000 [35]. Similar macro lenses that image only in the visible spectrum, such as the Canon EF-S 60 mm f/2.8, cost around GBP 270 [36], while a 60 mm f/2.0 Infrared (IR) lens, such as the LM60HC-IR—1” 5MP 60MM F2.0 C-MOUNT LENS costs about GBP 470 [37]. Sophisticated UV-transparent lenses are made of fused silica, quartz, calcium fluoride, magnesium fluoride, and sapphire. Fused silica has a transparency of 85% to 180 nm UV light at room temperature [38] and justifies our purpose of imaging in the UV-AB spectrum.

A wide-angle doublet lens of a 22 mm focal length with a relative aperture of f/1.6 was designed and developed using a positive meniscus element LE4173 (Thorlabs Inc., Newton, NJ, USA) and a biconvex element LB4854 (Thorlabs Inc., Newton, NJ, USA). The schematic of the lens design is shown in Figure 2. Doublet configuration of using a meniscus lens in conjunction with a biconvex lens rather than using a simple singlet biconvex lens improves performance at high relative apertures [39]. Also, this configuration controls the three aberrations that need to be addressed for monochromatic all-sky imaging: spherical aberration, astigmatism, and coma.

A 12 mm CS-mount lens for the Raspberry Pi IMX477 camera (Arducam, Nanjing, China) is stripped of its internal optical elements and spacers, and the lens body is used to house the two fused silica elements. Figure 3 shows the assembly of the two fused silica elements onto the 12 mm CS mount lens body. The 12 mm lens body provides a 25 mm distance between the positive meniscus and biconvex lenses.

A 25 mm diameter red rejection UV bandpass filter (XRR0340, Asahi Spectra, Tokyo, Japan) with centre wavelength (CWL) at 340 nm is placed between the C-mount lens holder and the DMK 41AU02.AS camera body. Figure 4 shows the UV bandpass filter assembled onto the camera body and the transmission curve of the UV bandpass filter.

### 2.3. Catadioptric System Assembly

The reflecting catadioptric system uses a twin-convex mirror assembly based on the reflecting all-sky camera designed by Michael Gadsden in 1974 [6]. Gadsden installed the reflecting all-sky imager at the Cromwell observatory in Kings College, University of Aberdeen. The current design for the UV all-sky imager is modelled based on the design data reported by Prof. Gadsden. The primary mirror used in the UV all-sky imager is a 30 cm acrylic convex mirror used for identifying blind spots in driveways, workplace depots, and factories. A 32 mm hole is laser-cut in the middle of the 30 cm acrylic convex mirror through which the camera lens projects onto the secondary mirror, an 8.5 cm convex mirror used for personal safety, and a rear deck view monitor. The 8.5 cm convex mirror is mounted in the interior at the midpoint of the 50 cm diameter acrylic dome. The schematic of the catadioptric system is shown in Figure 5, with the specifications of the mirrors and their position.

Using a twin-convex mirror system with a UV-transparent acrylic dome eliminates the use of a support frame structure to hold the camera, should a single convex mirror be used with the camera pointing towards it. The latter construction shall have the support frame structures visible in the generated all-sky image, obstructing the field of view of the sky. The distance of separation between the convex mirrors (D) decides the size of the supporting acrylic dome. A lower D leads to increased aberrations and restricts the field of view of the sky. Increasing the value of D requires a larger diameter acrylic dome; hence, the choice of D should be a trade-off between the design constraints and the camera focal length to cover the full sky.

The frame of the all-sky imaging system is constructed using 20 × 20 mm aluminium profiles, and the enclosure is laser-cut using 3 mm acrylic panels. The CAD model of the all-sky imager is shown in Figure 6. A 3D-printed convex mirror holder secures the secondary convex mirror to the centre of the acrylic dome.

Dew deposition on the acrylic dome and mirror affects the imaging system’s performance. In order to prevent this accumulation, the dome’s interior is subjected to a hot-air stream using two 12 V 100 W positive-temperature coefficient (PTC) fan heaters. Louvres are designed onto the supporting panel holding the large convex mirror and the acrylic dome to facilitate air movement. An Arduino© board controls the PTC heaters through a relay circuit. The heaters can be controlled remotely over the serial interface through a USB connector. Figure 7 shows the completed assembly of the all-sky imager deployed in an open field for capturing the all-sky image in UV-AB.

## 3. Results and Discussion

The modified ICX205AL sensor was validated for its performance in UV imaging. The effect of replacing the standard cover glass with the fused silica cover glass was analysed. A deuterium lamp, AvaLight-D-S (Avantes Inc., Lafayette, CO, USA), was a UV light source capable of generating UV light between 190 and 400 nm. The broadband UV light from the lamp was fed into a Newport Cornerstone 260 monochromator (Newport Corporation, Irvine, CA, USA) to produce a single spectral UV line. The output of the monochromator was fed into a trifurcated fibre optic cable of 600 µm cladding diameter, with each node connecting to a UV spectrometer (Avantes Inc., Lafayette, CO, USA), an 818-UV/DB low-power UV-enhanced silicon photodetector connected to 1936-R power meter (Newport Corporation, Irvine, CA, USA), and the COL-UV/VIS collimating lens (Avantes Inc., Lafayette, CO, USA). Figure 8 shows the spectral analysis setup. A similar setup was commissioned to calibrate the UV photodiodes of the HABIT (HabitAbility: Brine Irradiation and Temperature) environmental instrument for the ExoMars 2022 Surface Platform [41].

The collimating lens is housed inside a black 3D-printed cylindrical enclosure with the camera mounted to a custom-built focus assembly to ensure the collimated monochromatic UV beam is in the focus of the UV camera. The focus assembly is a simple mating spur gear assembly with a stepper motor controlled through an Arduino. The spectrometer and the power meter monitor the power output of the monochromatic beam and its wavelength.

The above setup performs the spectral analysis on the developed UV camera without the broadband UV filter before and after the cover glass change. The deuterium lamp is allowed to warm up for 30 min to ensure the output power remains stable. The deuterium lamp can generate 72 µW with the 600 µm cladding diameter fibre optic cable. The wavelength output of the monochromator is controlled through the MonoUT v5.1 software, beginning from 220 nm up to 420 nm in increments of 10 nm. The wavelength of the resulting monochromatic beam is monitored through the spectrometer, and the power meter records the beam’s intensity. Images of the beam are captured at 1 s exposure time. The average pixel readout value for every image captured in steps of 10 nm is calculated, and the resulting average pixel readout vs the wavelength is plotted, as shown in Figure 9. The average power output of the monochromatic UV beam between 220 and 420 nm is 22 nW.

We can observe that replacing the standard cover glass with a fused silica cover glass produces a slightly better performance of approximately 12% between 300 nm and 420 nm. The spectral performance of the camera with the red rejection UV bandpass filter mounted is analysed. Figure 10 shows the spectral response of the developed UV camera system in the near-UV region.

Comparative image analysis was performed with the developed UV camera using a standard optical glass lens and the developed fused silica lens, as shown in Figure 11. A 25 mm C-Mount Telephoto Lens for IMX477 Raspberry Pi High-Quality Camera (Waveshare, Shenzhen, China) made of optical glass was compared with the 22 mm C-mount fused silica lens. Both lenses have similar apertures, and the image was captured under similar lighting conditions at a constant gain setting. The standard glass lens has a very faint image observable only at an exposure of around 2 s, while the fused silica lens can capture a bright image at much faster speeds.

Figure 12 shows a passive UV all-sky image captured on a cloudy day. The image has a 130° field of view. However, there is cropping observed along the vertical axis. This cropping can be mitigated using a smaller primary convex mirror. The image is captured as 16-bit raw FITS data using the astronomical imaging software SharpCap. v4.0 The image is post-processed to remove the obstructions and isolate the sky alone. The image is then colour-masked with gradients based on the pixel brightness to distinguish cloud opacity. Accurate radiance calibration of the UV all-sky camera is essential for further scientific applications [42]. However, the passive imaging system can be used to study the spatial distribution and evolution of UV-detected elements.

The developed all-sky imager suffers from uncorrected astigmatism due to using two convex mirrors. This is a fundamental aberration native to catadioptric systems using two convex mirrors and can be corrected using deep-learning algorithms and optimization techniques [43]. The 50 cm acrylic dome does not provide a sufficient separation distance between the primary and the secondary convex mirror, leading to slight cropping observed in the image. This cropping can be mitigated by using a slightly higher-diameter acrylic dome. The sensitivity of the imaging system can be enhanced by binning multiple pixels into a single pixel at the cost of a lower resolution.

## 4. Conclusions

An all-sky imager for capturing a 130° field of view of the sky in the UV-AB spectrum has been developed, and its spectral response in the UV spectrum has been validated using monochromatic UV beams in 280–420 nm with an average power output of 22 nW. The developed UV all-sky imager is the first to map the sky in the UV spectrum and can find wide applications not restricted to research in a wide range of fields (atmosphere, vulcanism, meteors, …) but also to the defence sector, where monitoring the air space is crucial. The COTS-based approach exploited in developing the all-sky UV imager using easily accessible low-cost components has demonstrated an excellent cost-to-value ratio. The imaging system can be scaled to incorporate more sophisticated high-resolution cameras and lenses that significantly reduce optical aberrations. Accurate radiance calibration can scale the scientific applications of the developed UV all-sky imaging system for future investigation of cloud type and coverage detection and influence on solar irradiance in the UV spectrum.

## Figures and Tables

**Figure 1 sensors-23-07343-f001:**
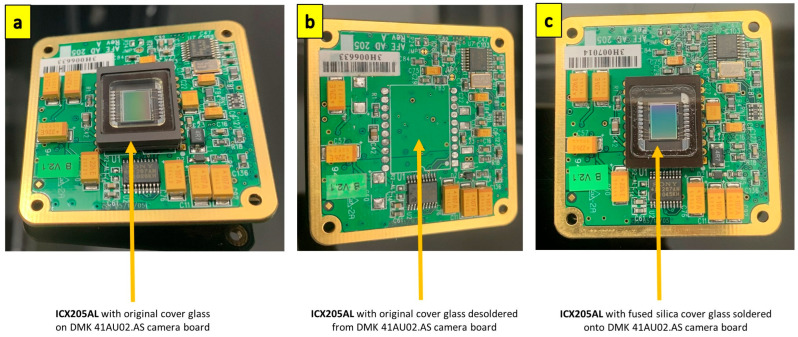
Replacement of standard cover glass ICX205AL sensor with fused silica cover glass ICX205AL sensor. (**a**) Standard cover glass ICX205AL sensor; (**b**) standard ICX205AL with standard cover glass de-soldered; (**c**) replacement ICX205AL with fused silica cover glass soldered back to the camera board.

**Figure 2 sensors-23-07343-f002:**
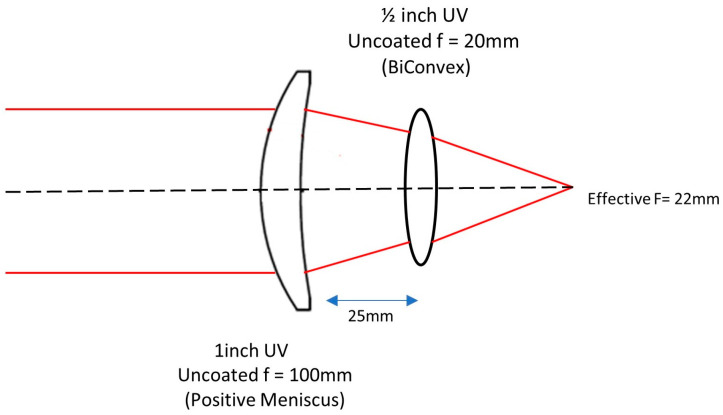
Schematic of the 22 mm fused silica lens designed from commercially available fused silica elements.

**Figure 3 sensors-23-07343-f003:**
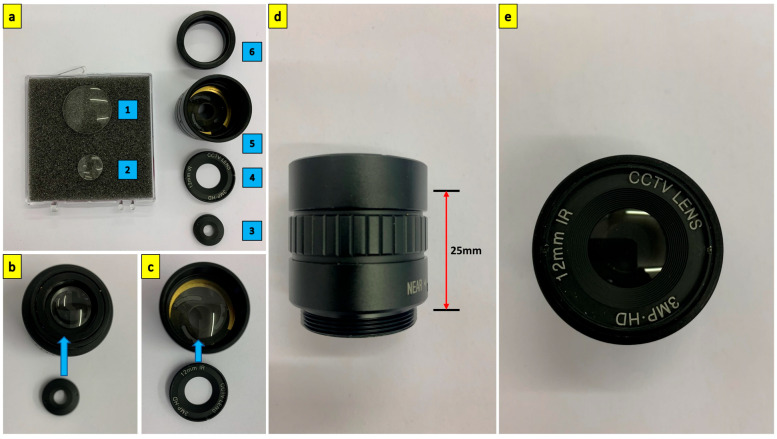
Assembly of the 22 mm fused silica lens. (**a**) Components of the fused silica lens: (1) 1-inch uncoated fused silica positive meniscus lens of focal length 100 mm; (2) ½-inch uncoated fused silica biconvex lens of focal length 20 mm; (3) rear optical element holder; (4) front optical element holder (optical stop); (5) 12 mm raspberry pi CS-mount lens body; (6) CS to C mount adapter. (**b**) Mounting rear element onto the lens body and securing with the rear optical element holder. (**c**) Mounting front element onto the lens body and securing with the front optical element holder. (**d**) The distance between the front (LE4173) and rear optical element (LB4854) is 25 mm. (**e**) Fully assembled 22 mm resulting focal length f/1.6 fused silica lens.

**Figure 4 sensors-23-07343-f004:**
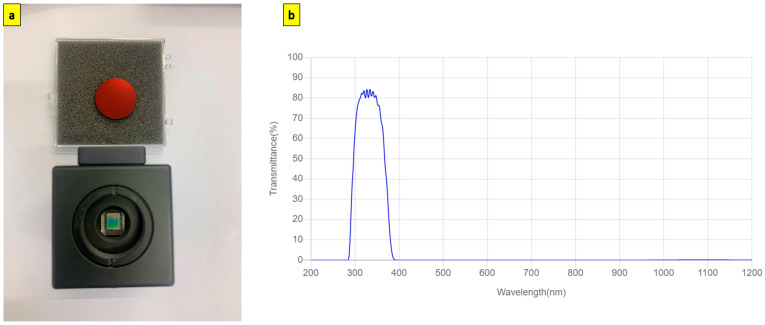
(**a**) Red rejection UV bandpass filter (XRR0340) mounted onto the fused silica cover glass DMK41AU02.AS camera (**b**) Transmission curve of XRR0340 red rejection UV bandpass filter [40].

**Figure 5 sensors-23-07343-f005:**
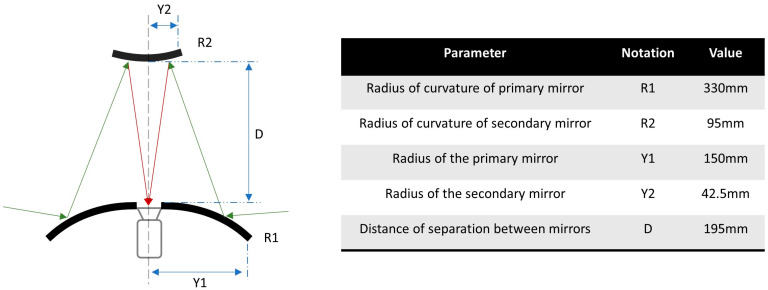
Schematic of the reflecting catadioptric system and the specifications of the mirrors used in the imager.

**Figure 6 sensors-23-07343-f006:**
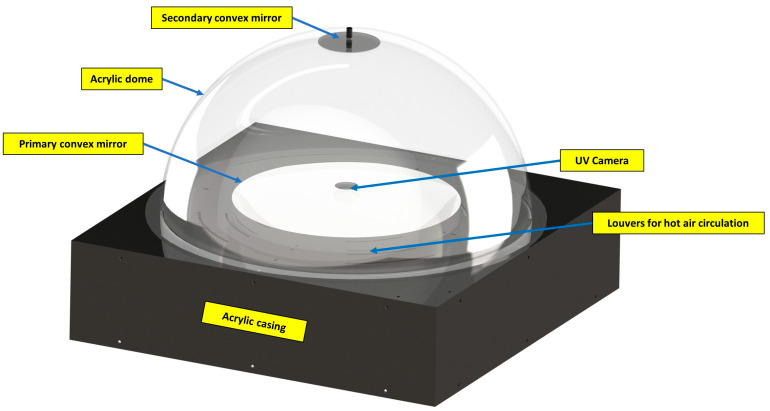
CAD model of the UV all-sky imager modelled in SolidWorks.

**Figure 7 sensors-23-07343-f007:**
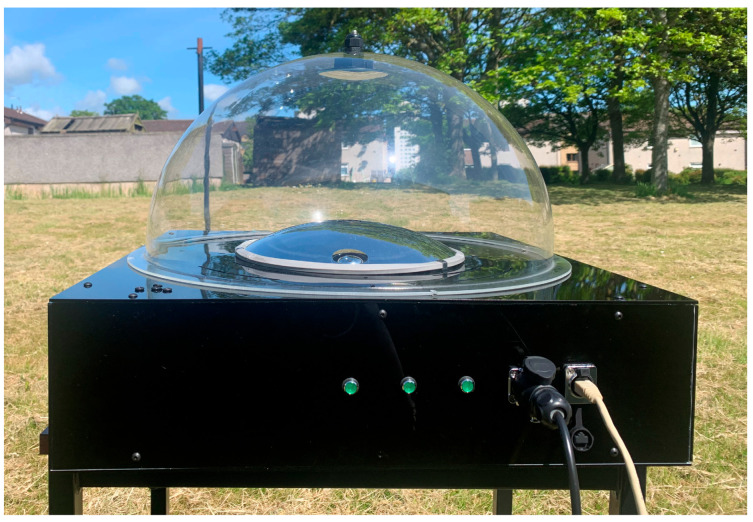
The assembled UV all-sky imager was deployed in the backyard.

**Figure 8 sensors-23-07343-f008:**
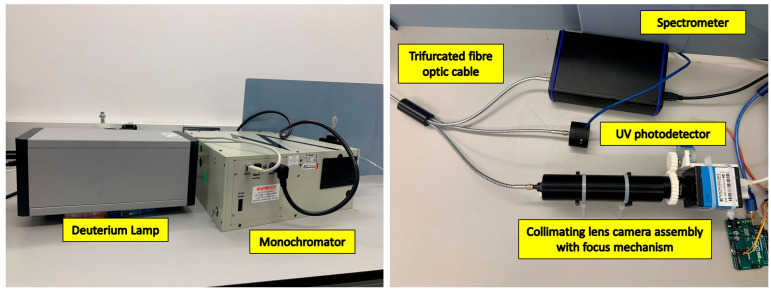
Spectral analysis setup to analyse the spectral response of the ICX205AL sensor before and after cover glass replacement.

**Figure 9 sensors-23-07343-f009:**
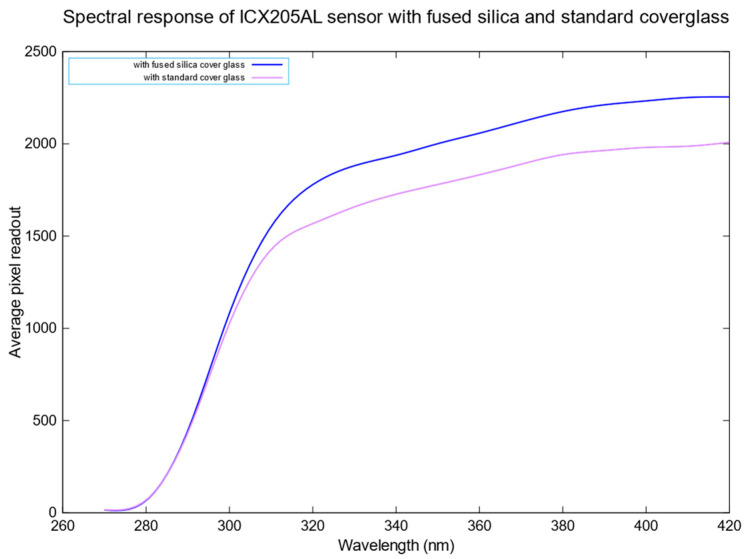
Spectral response of the camera sensor with and without cover glass. The experiments were conducted at constant room temperature with an average power of 22 nW collimated UV beam.

**Figure 10 sensors-23-07343-f010:**
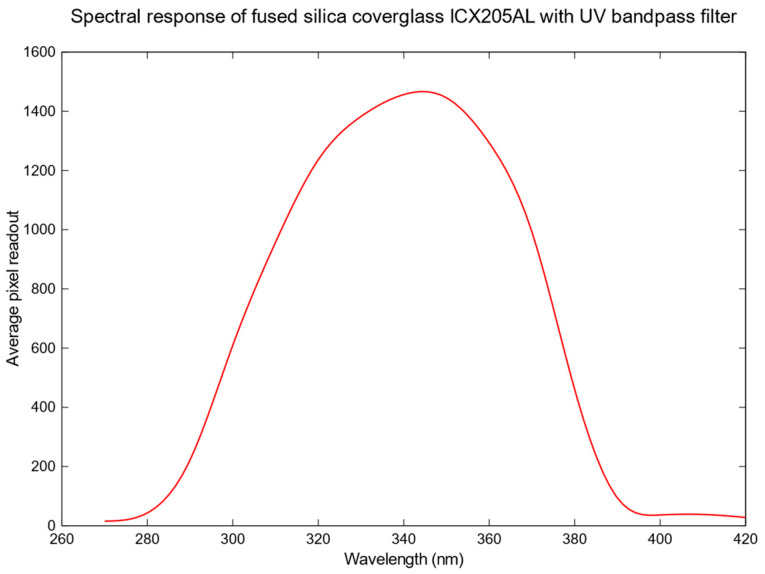
Spectral response of the fused silica cover glass ICX205AL with 340 nm CWL XRR red rejection UV bandpass filter.

**Figure 11 sensors-23-07343-f011:**
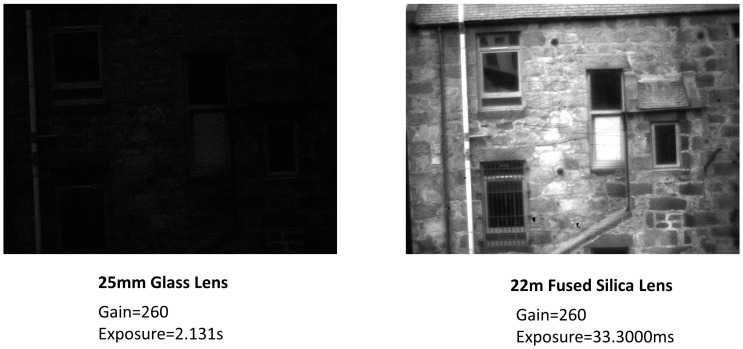
Comparison of images captured using the modified UV camera with a standard 25 mm glass lens against the developed 22 mm fused silica lens. The substantial improvement in signal output at a much lower exposure time using a fused silica lens is evident.

**Figure 12 sensors-23-07343-f012:**
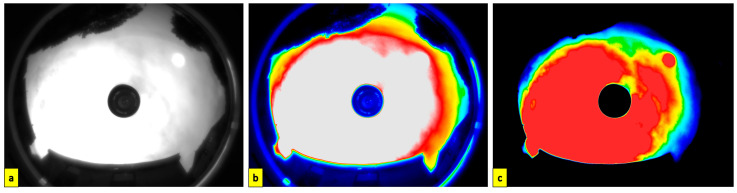
(**a**) All-sky image captured in UV-AB spectrum through the UV bandpass filter (280–380 nm) [Image captured at 57.09° N, 2.05° W, 11 m a.s.l on 2023-06-07T09_42_35]; (**b**) boundary detection to eliminate obstructions from all-sky image; (**c**) colour-masked image of the sky based on cloud opacity with red indicating lowest opacity and blue with highest opacity.

**Table 1 sensors-23-07343-t001:** Specification of ICX205AL sensor used in the UV all-sky imager.

Image Sensor Configuration	Interline Back-Illuminated CCD
Image size	8 mm diagonal (7.959 mm effective)
Total pixel count	1434 (H) × 1050 (V), 1.50 MP
Total effective pixel count	1392 (H) × 1040 (V), 1.45 MP
Total active pixel count	1360 (H) × 1024 (V), 1.40 MP
Chip size	7.60 mm (H) × 6.20 mm (V)
Pixel size	4.65 μm (H) × 4.65 μm (V)
Substrate material	Silicon

## Data Availability

Not applicable.

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
