# Peer review of "Design and Development of an Ultraviolet All-Sky Imaging System"

_sensors, 2023, doi:10.3390/s23177343_

Round 1
Reviewer 1 Report
This paper designed a UV all-sky imaging system which was based on low-cost, accessible, and scalable components. This system may be used in wide applications on astronomy, meteorology, atmospheric science, vulcanology, meteors and auroral monitoring, and the defence sector. However, the paper in its current state should not be published, unless the follows are addressed. See the comments below.
1. The purpose of the paper is to design a UV all-sky imaging system. However, the last part of the abstract describes the broad applications which were not investigated in this paper. So this part should be removed and the experimental results of the system should be added. Especially the radiance calibration.
2. Lines 120-131: Existing commercial UV sensitive sensors are limited in cost and availability, making them inconvenient to use. It is recommended to list several typical sensors for comparison and to explain why the ICX205AL sensor was chosen for this study.
3. In Part 2, you should first state the overall plan and explain why, and then describe the different parts of the plan. I think you should rewrite this part.
4. In addition to the design of the imager, the radiation calibration is particularly important, so the paper should pay more attention to the results of the instrument calibration. In particular, the radiation characteristics change with different viewing angles.
5. where is part 4?
Author Response
Dear Sir/Madam,
Thank you so much for your feedback and reviews. We have now addressed each point of the review. Please find below our response to the review.
- The purpose of the paper is to design a UV all-sky imaging system. However, the last part of the abstract describes the broad applications which were not investigated in this paper. So this part should be removed and the experimental results of the system should be added. Especially the radiance calibration.
Thank you for your feedback. We have now removed the broad application of the developed UV all-sky imagers in the abstract, considering that the paper focuses only on the development of the imaging system and not the applications that require calibration. The developed UV all-sky imager is a catadioptric system and is a noncentral system with possible misalignments between the camera and the reflective mirrors affecting the angular projection. This results in correction algorithms needed before radiance calibration can be performed. We aim to have another article published purely on this calibration process. We have mentioned now in the paper that the focus of the study is on the development of the UV all-sky imaging system and the radiance and angular calibration is not a part of this study. We have added reference to three algorithms that can be utilized for such noncentral imaging systems to correct the angular projection. We have retained the summary of the applications of the developed UV all-sky imagers in the Introduction section in lines 70-90 to give an insight to the readers on the fields it can be used for.
- Lines 120-131: Existing commercial UV sensitive sensors are limited in cost and availability, making them inconvenient to use. It is recommended to list several typical sensors for comparison and to explain why the ICX205AL sensor was chosen for this study.
Thank you for your comment. ICX205AL sensor was specifically chosen for the UV camera development as it belongs to the family of HAD (Hole-Accumulation Diode) sensors which offer high sensitivity and low dark current required for all-sky imaging. The heritage of ICX205AL sensors in all-sky imaging studies of the night sky to monitor atmospheric turbulence [1], [2] justified its imaging performance at low signal conditions. We have now included the explanation of using ICX205AL sensor in lines 128-132 with two references added to the discussion.
- In Part 2, you should first state the overall plan and explain why, and then describe the different parts of the plan. I think you should rewrite this part.
Thank you for your feedback. We have now included a short summary of the overall plan in this section in lines 113-117.
- In addition to the design of the imager, the radiation calibration is particularly important, so the paper should pay more attention to the results of the instrument calibration. In particular, the radiation characteristics change with different viewing angles.
Thank you for your feedback. We have addressed the feedback collectively in point 1.
- where is part 4?
Thank you for your feedback. We have corrected the section numbering in the manuscript.
References:
[1] Aziz Ziad, Julien Chabé, Yan Fanteï-Caujolle, Eric Aristidi, Catherine Renaud. CATS: a new station for a complete characterization of atmospheric turbulence. AO4ELT5, 2017, Tenerife, Spain. Available at: https://hal.science/hal-02477428/
[2] Julien Chabé, Aziz Ziad, Yan Fantéï-Caujolle, Éric Aristidi, Catherine Renaud, Flavien Blary, and Mohammed Marjani "The Calern atmospheric turbulence station", Proc. SPIE 9906, Ground-based and Airborne Telescopes VI, 99064Z (27 July 2016); Available at: https://doi.org/10.1117/12.2232383
Reviewer 2 Report
The manuscript begins by introducing the previous all-sky imaging systems and proposing the issue that the existing “all-sky imaging cameras operate in the visible and infrared regions”. To address this issue, the authors develop an all-sky imager capable of capturing 130° wide-angle sky images in the UV-AB spectrum. The UV all-sky imaging system is constructed with low-cost, accessible, and scalable components. The imager includes three parts: a UV camera sensor, a UV transparent lens and the catadioptric system assembly. They replace the cover glass of a commercial monochrome sensor to improve its responsivity to the UV spectrum. They assemble a UV lens with commercially available elements and a commercial lens body. They design a catadioptric reflecting system using commercially available convex mirrors. The imaging performance in the UV spectrum of the imager is validated and the results are discussed. A brief conclusion closes the paper.
The paper is logically organized and very easy to read. Authors provide sufficient details and explicit descriptions in the manuscript. The English is well-written. The work in this manuscript is complete and referentially valuable for similar engineering practices with requirements on cost, accessibility and scalability. But it seems that creativity is somewhat insufficient.
I think revisions are required before the paper is published:
-line 62: The authors mentioned “an imaging system does not exist to provide images in the Ultraviolet (UV) spectrum domain.”, but is it really the case? If this is really the case, since that “The all-sky camera operating in the UV-AB region is an excellent tool for studying the enhancement of solar radiation in the UV region due to radiation scattering from the clouds”, I think the authors need to explain why there’s no existing UV imaging system. The innovation of the author’s work can also be highlighted with the explanation.
-The authors mention the low cost of their system several times in the manuscript. However, the paper does not give a cost comparison with other commercial systems. I think the paper will benefit from a comparison of costs.
-Fig.1: I noticed that an electronic component is missing from the upper left corner of Fig. 1b. Does this mean that the authors also modified the circuit board? Explanations are needed.
-Section.2.2: The authors use a commercial lens body. How is the spacing between the fused silica lenses inside the lens controlled?
-line 181: It should be “(6) 5mm spacer (CS to C mount)”.
-Please consider adding analysis for the aberrations of the imager system.
-Since that many all-sky imaging applications utilize the polarization of the UV light, does the system affect the polarization state?
Author Response
Dear Sir/Madam,
Thank you so much for your feedback and reviews. We have now addressed each point in the review. Please, find below the response to the review.
-line 62: The authors mentioned “an imaging system does not exist to provide images in the Ultraviolet (UV) spectrum domain.”, but is it really the case? If this is really the case, since that “The all-sky camera operating in the UV-AB region is an excellent tool for studying the enhancement of solar radiation in the UV region due to radiation scattering from the clouds”, I think the authors need to explain why there’s no existing UV imaging system. The innovation of the author’s work can also be highlighted with the explanation.
Thank you for your feedback. We have now addressed the reason for the absence of such UV all-sky imagers in lines 58-63 and the innovativeness of using a catadioptric system to circumvent the issue of the lack of fisheye lens constructed from UV transparent materials.
-The authors mention the low cost of their system several times in the manuscript. However, the paper does not give a cost comparison with other commercial systems. I think the paper will benefit from a comparison of costs.
Thank you for your feedback. There exists no such commercial system to image the full sky in the UV spectrum and we have undertaken a low-cost approach of designing a catadioptric system rather than the development of a fused silica fisheye lens which is quite expensive. We have mentioned low-cost several times in the paper to stress to the readers that cost-effective designs can be exploited to achieve the results with limited funding availability.
-Fig.1: I noticed that an electronic component is missing from the upper left corner of Fig. 1b. Does this mean that the authors also modified the circuit board? Explanations are needed.
Thank you for your query. The electronic components were removed to aid in desoldering the imaging sensor from the circuit board and they were soldered back as shown in Fig1C.
-Section.2.2: The authors use a commercial lens body. How is the spacing between the fused silica lenses inside the lens controlled?
Thank you for your query. The commercial lens body used is a 12mm C-mount lens. This 12mm lens is constructed of 4 optical elements spaced apart using spacers inside the assembly. We have removed all the optical elements and the spacers and used just the frame that has a 25mm separation distance between the front and rear lens holder. We have now mentioned that the spacers were also removed in line 170.
-line 181: It should be “(6) 5mm spacer (CS to C mount)”.
Thank you for the feedback. We have now corrected line 178 of the manuscript.
-Please consider adding analysis for the aberrations of the imager system.
Thank you for your feedback. The developed UV all-sky imager is a catadioptric system and is a noncentral system with possible misalignments between the camera and the reflective mirrors affecting the angular projection. There exist algorithms such as the blackbox model [1], complete model [2] and generalized union model [3] that can correct the angular projection minimizing the aberrations. We aim to have a separate paper on the calibration process and hence we have not taken the calibration into account in this paper and focussed only on the development of the camera.
-Since that many all-sky imaging applications utilize the polarization of the UV light, does the system affect the polarization state?
Thank you for your query. We have not considered polarization study in the current study. It is really interesting to analyse the polarization in the UV all-sky imager, which after calibration we would like to include in a separate article.
References:
[1] Grossberg, M.D. and Nayar, S.K. (2005) ‘The RAXEL imaging model and Ray-based calibration’, International Journal of Computer Vision, 61(2), pp. 119–137. Available at: https://doi.org/10.1023/b:visi.0000043754.56350.10
[2] Gonçalves, N. and Araújo, H. (2009) ‘Estimating parameters of noncentral catadioptric systems using bundle adjustment’, Computer Vision and Image Understanding, 113(1), pp. 11–28. Available at: https://doi.org/10.1016/j.cviu.2008.06.004
[3] Xiang, Z., Dai, X. and Gong, X. (2013) ‘Noncentral catadioptric camera calibration using a generalized unified model’, Optics Letters, 38(9), p. 1367. Available at: https://doi.org/10.1364/ol.38.001367
Round 2
Reviewer 1 Report
1. Line 245: figure 1 should be figure 6.
2. You claimed that the performance had been validated. the conclusion of the validation should be added to the abstract and the conclusions.
3. The author claimed that the paper details on the development of the imaging system and does not cover the radiance and angular calibration of the imaging system, which will be detailed in a separate article. How do the readers understand the performance of the system?
Author Response
Dear Sir/Madam,
Thank you so much again for your feedback and reviews. We have now addressed each point of the second review. Please find below our response to the review.
- Line 245: figure 1 should be figure 6.
Thank you for your feedback. Now we have labelled it as Figure 6.
- You claimed that the performance had been validated. the conclusion of the validation should be added to the abstract and the conclusions.
Thank you for your feedback. We have now mentioned in both the abstract (lines 16-18) and conclusion (lines 355-356) the spectral response validation of the camera system in the 280-420nm UV wavelength range using a monochromatic UV beam with an average power output of 22nW.
- The author claimed that the paper details on the development of the imaging system and does not cover the radiance and angular calibration of the imaging system, which will be detailed in a separate article. How do the readers understand the performance of the system?
Thank you for your feedback. In the scope of the current work, we present the readers the design of a UV all-sky imaging system using catadioptrics, where we focus only on the spectral responsivity performance of the system to UV-AB spectrum (280-420nm) which is crucial for passive all-sky imaging of studying the spatial distribution and evolution of UV detected elements. With radiance and angular calibration, the scope of the developed UV all-sky imaging system can be extended to more scientific applications than passive imaging wherein readers can use various filters (narrowband or wideband) to study the all-sky in specific UV wavelengths.
Reviewer 2 Report
The authors have addressed most of my concerns.
1. As to the motivation of developing a low-cost system, maybe the authors could compare the cost of each component in a UV camera and a visible/infrared camera to analyze the reason why no such commercial system exists to imaging the full sky in UV spectrum.
2. As to the aberrations of the system, I suggest the authors at least state the aberration requirements for an all-sky imaging system aiming for the applications including atmospheric science etc.
3. Figure 1 in line 245 should be Figure 6.
Author Response
Dear Sir/Madam,
Thank you so much for your feedback and reviews. We have now addressed each point in the second review. Please, find below the response to the review.
- As to the motivation of developing a low-cost system, maybe the authors could compare the cost of each component in a UV camera and a visible/infrared camera to analyze the reason why no such commercial system exists to imaging the full sky in UV spectrum.
Thank you for your feedback. We have now included in lines 136-139, the comparative cost of a commercial UV sensitive sensor such as the Sony IMX487 [1] of £10,000 against a similar specification sensor capturing in visible spectrum such as the Sony IMX546 [2] that costs only around £1000. Similarly, in lines 182-191 we have now made a cost comparison between the commercial UV grade lens such as the Jenoptik 60mm F/4 Apochromatic (APO) lens [3], the Nikkor 105mm F/4.5 [4] costing between £7000 to £12000 against macro lenses that capture in visible and infrared such as the Canon EF-S 60mm f/2.8 [5] and LM60HC-IR - 1" 5MP 60MM F2.0 [6] respectively costing between £270 to £470. We have now mentioned that the limited availability of the UV grade lens and sensors and their high cost are the major limiting factors in the development of a commercial UV all-sky imager.
- As to the aberrations of the system, I suggest the authors at least state the aberration requirements for an all-sky imaging system aiming for the applications including atmospheric science etc.
Thank you for your feedback. We have now included the details of all three aberrations (lines 201-202) that need to be addressed for all-sky UV imaging namely spherical aberration, astigmatism, and coma. The doublet lens configuration solves most of the aberrations, but the use of a catadioptric system with two convex mirrors leads to uncorrected astigmatism. This aberration is native to such systems and now we have included in lines 345-347 about mitigating astigmatism using deep learning and optimization techniques [7].
- Figure 1 in line 245 should be Figure 6.
Thank you for your feedback. Now we have labelled it as Figure 6.
References:
[1] Sony IMX487 Ultraviolet Scientific Grade Camera - XIMEA. Available at: https://www.ximea.com/en/products/xilab-application-specific-custom-oem/sony-imx487-ultraviolet-scientific-grade-camera?responsivizer_template=desktopltraviolet+scientific+grade+camera (Accessed: 15 August 2023).
[2] Lucid vision labs tritonTM tri081s-cc, Sony IMX546, 8.1MP, Color Camera - Edmund Optics Worldwide. Available at: https://www.edmundoptics.co.uk/p/lucid-vision-labs-tritont-tri081s-cc-sony-imx546-81mp-color-camera/45507/ (Accessed: 15 August 2023).
[3] LDP LLC - UV Lenses. Available at: https://maxmax.com/uv-lenses (Accessed: 15 August 2023).
[4] 105mm f/4.5 UV nikkor AIS (2022) Grays of Westminster Online Shop. Available at: https://shop.graysofwestminster.co.uk/product/105mm-f-4-5-uv-nikkor-ais (Accessed: 15 August 2023).
[5] Canon EF-S 60mm F2.8 USM MACRO A3061303 - Aperture UK. Available at: https://www.apertureuk.com/canon-ef-s-60mm-f28-usm-macro-a3061303.html (Accessed: 15 August 2023).
[6] LM60HC-IR - 1" 5MP 60mm F2.0 C-mount lens - Torchlight Vision. Available at: https://www.torchlightvision.com/en-gb/products/lm60hc-ir (Accessed: 15 August 2023).
[7] Tien, C.-L., Chiang, C.-Y. and Sun, W.-S. (2022) ‘Design of a miniaturized wide-angle fisheye lens based on deep learning and Optimization Techniques’, Micromachines, 13(9), p. 1409. Available at: https://doi.org/10.3390/mi13091409